# Appetite controlled by a cholecystokinin nucleus of the solitary tract to hypothalamus neurocircuit

**Giuseppe D'Agostino[1,2]\*, David J Lyons[1], Claudia Cristiano[1], Luke K Burke[2], Joseph C Madara[3], John N Campbell[3], Ana Paula Garcia[2], Benjamin B Land[4], Bradford B Lowell[3], Ralph J Dileone[4], Lora K Heisler[1,2]\***

[1]Rowett Institute of Nutrition and Health, University of Aberdeen, Aberdeen, United Kingdom; [2]Department of Pharmacology, University of Cambridge, Cambridge, United Kingdom; [3]Division of Endocrinology, Diabetes and Metabolism, Department of Medicine, Beth Israel Deaconess Medical Center, Harvard Medical School, Boston, United States; [4]Department of Psychiatry, Yale University School of Medicine, New Haven, United States

**Abstract** The nucleus of the solitary tract (NTS) is a key gateway for meal-related signals entering the brain from the periphery. However, the chemical mediators crucial to this process have not been fully elucidated. We reveal that a subset of NTS neurons containing cholecystokinin (CCK[NTS]) is responsive to nutritional state and that their activation reduces appetite and body weight in mice. Cell-specific anterograde tracing revealed that CCK[NTS] neurons provide a distinctive innervation of the paraventricular nucleus of the hypothalamus (PVH), with fibers and varicosities in close apposition to a subset of melanocortin-4 receptor (MC4R[PVH]) cells, which are also responsive to CCK. Optogenetic activation of CCK[NTS] axon terminals within the PVH reveal the satiating function of CCK[NTS] neurons to be mediated by a CCK[NTS]→PVH pathway that also encodes positive valence. These data identify the functional significance of CCK[NTS] neurons and reveal a sufficient and discrete NTS to hypothalamus circuit controlling appetite.

\*For correspondence: giuseppe. dagostino@abdn.ac.uk (GD'Agostino); lora.heisler@abdn. ac.uk (LKH)

**Competing interests:** The authors declare that no competing interests exist.

## Introduction

Obesity has emerged as one of the global healthcare challenges of the 21st century. Common obesity is primarily a consequence of food intake beyond the body's energetic requirements, with the excess energy consequently stored as fat. Mechanistically, the brain integrates and responds to multiple homeostatic hormones, neurotransmitters, nutrients and peripherally generated neural signals to maintain energetic balance (*Dietrich and Horvath, 2013*; *Morton et al., 2014*). One of the primary integration nodes within the brain for meal-related and metabolic signals from the periphery is the nucleus of the solitary tract (NTS) (*Grill and Hayes, 2012*; *Rinaman, 2010*; *Schwartz, 2010*; *Wu et al., 2012*). The NTS hosts a variety of factors to control homeostatic functions, including a small subset of neurons that express cholecystokinin (CCK[NTS]) (*Garfield et al., 2012*; *Herbert and Saper, 1990*; *Vitale et al., 1991*).

CCK consists of a family of peptides, the best characterized of which is a 33 amino acid peptide secreted from endocrine cells in the jejunum in response to nutrients in the intestinal lumen. This gut-derived peptide has a number of gastrointestinal (GI) functions including the promotion of satiation/satiety (*Gibbs et al., 1973*; *Saito et al., 1981*; *Smith and Gibbs, 1994*). Peripherally derived CCK does not readily penetrate the brain (*Fan et al., 1997*; *Passaro et al., 1982*). However, its short-term effect on appetite has been in part attributed to stimulation of vagal sensory neurons

**eLife digest** Obesity primarily results from eating more food than the body requires, the energy from which is then stored as fat. In recent years obesity has become increasingly common, with the resulting health problems presenting one of the major healthcare challenges of the twenty-first century. New ways to tackle the obesity epidemic are therefore required to improve human health on a global scale.

To regulate how much food is eaten, the gut sends chemical messengers to the brain about how much food has been consumed. These messengers activate particular cells in the brain that signal to other brain regions to trigger a decision about whether we've had enough food to eat. This raises a question: if we can artificially activate these cells, can we 'trick' the brain into thinking that food has been consumed?

A brain region called the nucleus of the solitary tract (NTS) is known to play a key role in receiving signals from the gut about meals. By studying mice, D'Agostino et al. found that cells in the NTS that make a brain hormone called cholecystokinin (CCK) are particularly activated by food.

Further experiments then used a technique called optogenetics to activate these cells in mice that had free access to different types of food. This activation significantly reduced how hungry the mice were, causing them to eat less food and lose weight. D'Agostino et al. also showed that CCK cells relay the signal about food intake to a brain region called the hypothalamus.

Overall, D'Agostino et al. have found a way to trick the brain into thinking that food has been eaten when it actually hasn't, and for this reason mice eat less without feeling hungry and lose weight. The next step is to try and find a way to activate the CCK cells in obese humans who have health complications associated with excess body weight.

influencing the brainstem (*Fan et al., 1997*; *Gibbs et al., 1973*; *Saito et al., 1981*; *Smith and Gibbs, 1994*).

CCK is also synthesized within the brain, where it is post-translationally processed into an 8 amino acid peptide (CCK-8) that reduces food intake when centrally infused (*Blevins et al., 2000*; *Hirosue et al., 1993*). However, the source of brain CCK that controls appetite has not been established. Recently developed chemogenetic and optogenetic approaches now provide a means to decipher the real-time contribution of discrete neuronal populations and networks to behavior (*Sternson and Roth, 2014*; *Tye and Deisseroth, 2012*), although, no direct functional assessment of CCK[NTS] neurons has yet been undertaken.

## Results and discussion

### CCK[NTS] neurons are responsive to nutritional state

Although CCK is produced in the NTS (*Garfield et al., 2012*; *Herbert and Saper, 1990*), little is known about the function of this discrete source of brain CCK. To facilitate visualization and characterization of CCK[NTS], we employed a knock-in mouse line expressing Cre recombinase at the *Cck* locus (*Cck*-iCre) crossed with a Cre-dependent enhanced yellow fluorescent protein reporter (Ai3) line (*Cck[YFP]*; *Figure 1A*). CCK-eYFP cells were most abundantly expressed within the caudal aspect of the NTS (*Figure 1B,C*), a brain region innervated by gastrointestinal vagal afferents and involved in nutrient sensing (*Appleyard et al., 2005*; *Blouet and Schwartz, 2012*; *Ritter, 2011*; *Schwarz et al., 2010*).

To determine whether CCK[NTS] cells are responsive to food intake, *Cck[YFP]* mice were exposed to either *ad libitum* fed, dark cycle fasted or dark cycle fasted followed by 2 hr re-feeding conditions, and a surrogate marker of neuronal activation, c-Fos immunoreactivity (IR), was assessed. In contrast to light cycle *ad libitum* fed and the dark cycle fasted conditions, ingestion of food on an empty stomach significantly increased c-Fos-IR within CCK[NTS] cells, indicating responsiveness to food consumption (*Figure 1D,E*). To clarify whether this response is related to the nutrients, as opposed to stomach stretching or orosensory aspects of feeding, we next investigated whether CCK[NTS] cells are responsive to nutrients if directly delivered to the stomach. Dark cycle fasted *Cck[YFP]* mice were

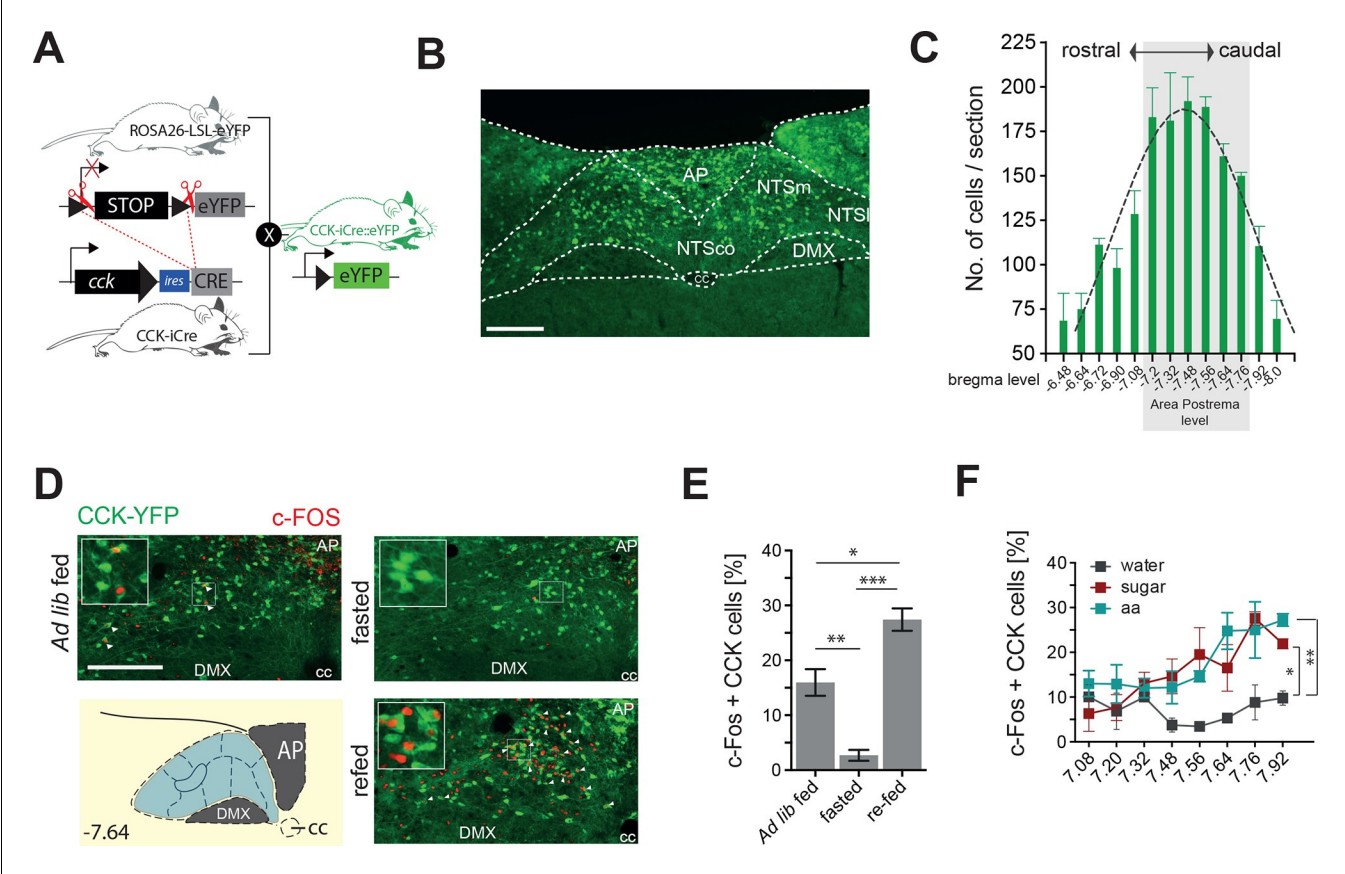

**Figure 1.** CCK[NTS] neurons are activated by feeding. (**A**) Schematic of mouse crossing to generate a *CCK-iCre::R26-loxSTOPlox-eYFP* mouse line (*Cck*[YFP]). (**B**) Representative expression of *Cck*[YPF] in a NTS coronal section. (**C**) Quantification of *Cck*[YPF]-expressing cells across the rostral-to-caudal extent of the NTS. (**D**) Representative c-Fos-IR in *Cck*[YPF] NTS cells in *ad libitum* fed, fasted or fasted then re-fed mice (white arrows denote colocalized neurons) and (**E**) quantification of c-FOS-positive *Cck*[YPF] NTS cells (n = 3–4; one-way ANOVA $F_{(2,7)}$ = 39.82, p = 0.0001; Sidak's *post hoc* comparison). (**F**) Quantification of c-FOS-IR across the rostral-to-caudal extent of the NTS in *Cck*[YPF] cells by bregma level following intragastric delivery of water, amino acids (aa) or sucrose (n = 3–5 per group; one-way ANOVA, $F_{(2,21)}$ = 7.280, p = 0.0040; Tukey's *post hoc* comparison). *p<0.05, **p< 0.01, ***p< 0.001. Scale bar in **B** and **D** represents 200 µm. AP, area postrema; DMX, dorsal motor nucleus; NTSco, nucleus of the solitary tract, commissural part; NTSm, nucleus of the solitary tract, medial part; NTSl, nucleus of the solitary tract, lateral part.

intragrastrically delivered isovolumetric (0.5 ml) non-nutritive water or isocaloric (1 kcal) sucrose or amino acids. As observed with chow intake, gavage of sucrose or amino acids significantly increased c-Fos-IR within CCK[NTS] cells compared with water (*Figure 1F*). These results suggest that CCK[NTS] cells are activated by nutrient intake.

## CCK[NTS] neuron activation reduces food intake and body weight

We next considered whether activation of CCK[NTS] neurons could promote satiety by communicating a nutrient consumption signal. *Cck*-iCre mice were bilaterally injected into the NTS with AAVs that mediate the Cre-dependent expression of designer receptors exclusively activated by designer drugs (DREADDs; expressed as DREADD-mCherry fusion proteins, hM3Dq; *Cck*-iCre-hM3D$_q$-mCherry[NTS]) (*Figure 2A*). DREADDs are designer muscarinic receptor variants that can only be activated by an otherwise biologically inert designer drug, clozapine-N-oxide (CNO) (*Alexander et al., 2009*; *Krashes et al., 2011*). Using this approach, we achieved expression of the DREADD-fused mCherry reporter protein in the caudal aspect of the NTS (*Figure 2B*). CNO activated *Cck*-iCre-hM3D$_q$-mCherry[NTS] cells *in vivo* as shown by increased c-Fos-IR (*Figure 2C*) and *ex vivo* using electrophysiology in NTS slices (*Figure 2D*). Activation of CCK[NTS] neurons in *ad libitum* fed *Cck*-iCre-hM3D$_q$-

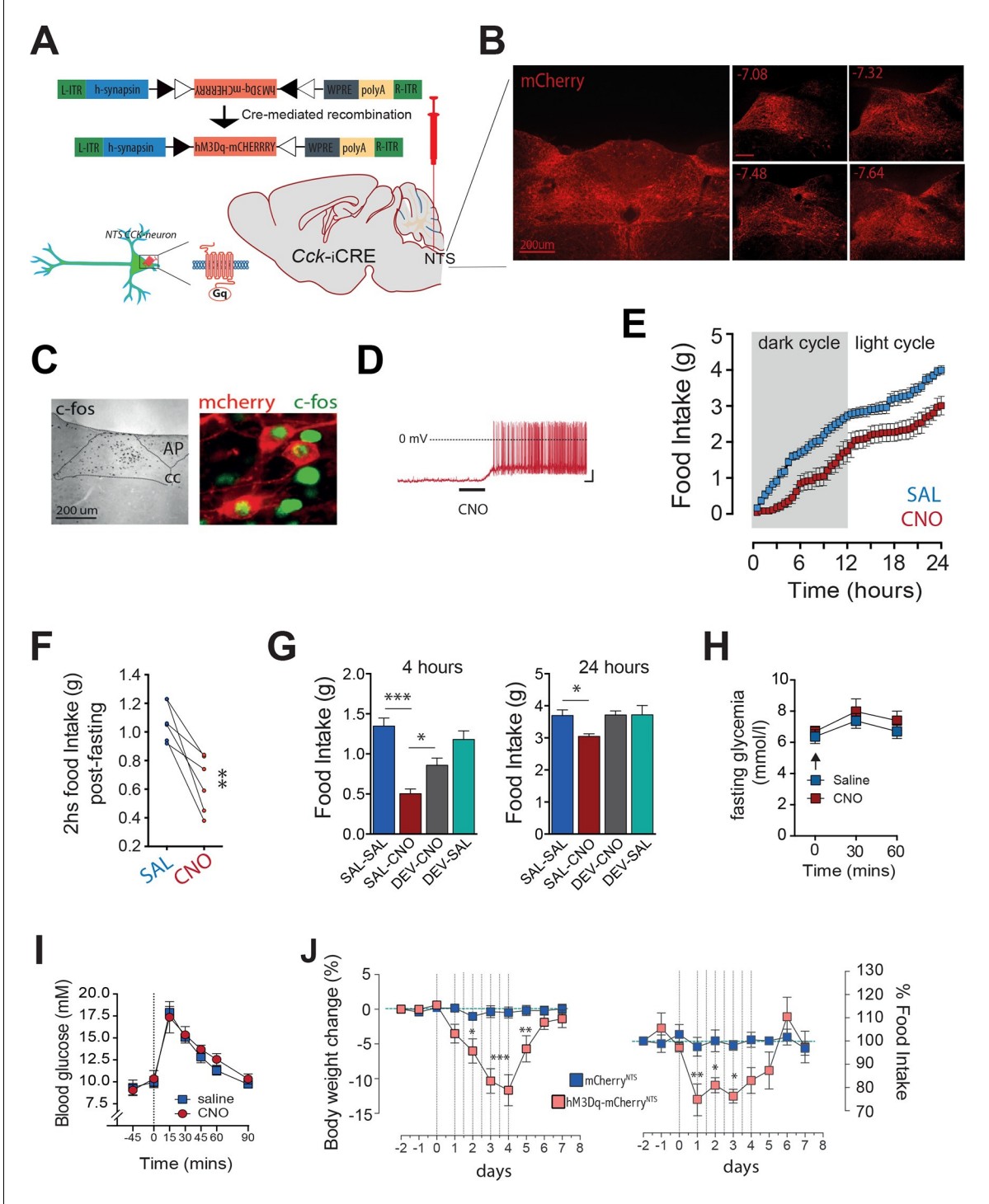

**Figure 2.** Activation of CCK[NTS] neurons reduces feeding and body weight. (A–D) Bilateral stereotaxic injection of Cre-dependent excitatory hM3D$_q$-mCherry virus into the NTS of male *Cck-iCre* mice facilitated activation of CCK[NTS] neurons. (A) Schematic and Cre-mediated recombination of hM3D$_q$-mCherry allele. (B) Representative image of Cre-dependent expression of hM3D$_q$-mCherry within the caudal aspect of the NTS of a *Cck-iCre* mouse (coronal sections; numbers indicate bregma levels, scale bar represents 200 μm). (C) c-Fos-IR in the NTS and co-expression (green) in hM3D$_q$-mCherry-transduced CCK[NTS] neurons (red) (scale bar represents 200 μm). (D) Membrane potential and firing rate of *Cck-iCre*-hM3D$_q$-mCherry[NTS] neurons increased upon 5 μM CNO application. (E) *Cck-iCre*-hM3D$_q$-mCherry[NTS] mice exhibited a significant reduction of spontaneous feeding following CNO, compared to saline, administration (n = 6; RM ANOVA, main effect of treatment [$F_{(1,5)}$ = 22.41, p = 0.0052], main effect of time [$F_{(47,235)}$ = 101.6, p<0.0001], and interaction [$F_{(47,235)}$ = 1.807, p = 0.0023]); tick marks on x-axis represent 3 hr, measurements collected with 30-min intervals) and (F) a reduction of post-fast re-feeding following CNO compared to saline administration (n = 6; paired t test, $t_{(5)}$ = 4.769, p = 0.005). (G) CNO-induced

*Figure 2 continued on next page*

*Figure 2 continued*

reduction in spontaneous feeding was attenuated by pre-treatment with the CCK-A-receptor antagonist, devazepide (DEV; 1 mg/kg), (n = 6; ANOVA, $F_{(3,20)}$ = 16.81, p<0.0001; Sidak's *post hoc* comparison \*\*\*p<0.001, \*p< 0.05). (H) CNO did not change fasting glucose level or (I) glucose disposal rate following a systemic glucose load (1 g/kg, IP). (J) Repeated CNO administrations over 4 days reduced body weight (n = 6; RM ANOVA, main effect of treatment [$F_{(1,100)}$ = 60.78, p<0.0001], main effect of time [$F_{(9,100)}$ = 8.877, p<0.0001], and interaction [$F_{(9,100)}$ = 7.483, p<0.0001]) and decreased food intake (main effect of treatment [$F_{(1,100)}$ = 16.13, p = 0.0001], main effect of time [$F_{(9,100)}$ = 4.106, p = 0.0002], and interaction [$F_{(9,100)}$ = 3.307, p = 0.0014]; Sidak's *post hoc* comparisons, \*p<0.05, \*\*p<0.01, \*\*\*p<0.001) in *Cck-iCre*-hM3D$_q$-mCherry[NTS] as compared to *Cck-iCre*-mCherry[NTS] mice.

mCherry[NTS] mice injected with CNO (0.3 mg kg$^{-1}$, IP) suppressed food intake for 24 hr (*Figure 2E*). Likewise, CCK[NTS] neuron activation suppressed re-feeding in food deprived mice (*Figure 2F*).

We next tested whether CCK itself mediates these food supressing effects of CCK[NTS] neuron activation. *Cck-iCre*-hM3D$_q$-mCherry[NTS] mice were pre-treated with the CCK-A receptor antagonist devazepide which blunted CNO-induced appetite suppression both 4 hr and 24 hr after treatment (*Figure 2G*), with no effects in control mice. These results demonstrate CCK-A receptors as downstream effectors of CCK[NTS] neuron-mediated reduction in feeding.

Given our recent findings that parabrachial CCKergic (CCK[PBN]) transmission regulates hepatic glucose production (*Flak et al., 2014*; *Garfield et al., 2014*), we next investigated whether CCK[NTS] neurons impact glucose homeostasis. Unlike CCK[PBN] neurons, chemogenetic activation of CCK[NTS] neurons in *Cck-iCre*-hM3D$_q$-mCherry[NTS] mice altered neither circulating blood glucose levels (*Figure 2H*) nor glucose disposal following a systemic glucose load (*Figure 2I*). These results reveal a neuroanatomical functional divergence in the energy-intake- versus energy-metabolic features of CCKergic neurotransmission within the brain, with CCK[NTS] neurons modulating food intake, but not glucose homeostasis.

We next tested whether long-term stimulation of CCK[NTS] neurons promotes a sustained reduction in food intake and body weight loss. We treated *Cck-iCre*-hM3D$_q$-mCherry[NTS] and control *Cck-iCre*-mCherry[NTS] mice with CNO for 4 days (two injections per day). Repeated activation of CCK[NTS] neurons resulted in a pronounced reduction in food intake with a concomitant reduction in body weight (*Figure 2J*). Following withdrawal of CNO treatment, mice returned to the original body weight within approximately 48 hr and to baseline food intake after a transient phase of rebound feeding (*Figure 2J*). These data demonstrate that sustained activation of CCK[NTS] neurons is sufficient to promote pronounced and reversible anorexia and body weight loss in mice.

Together, these data reveal that activation of CCK[NTS] neurons promotes anorexia and weight loss, attenuating the homeostatic drive to eat even in face of marked negative energy balance (as occurs in food deprivation).

## A CCK[NTS]→PVH circuit controls appetite

As a first step to delineate the circuits through which CCK[NTS] neurons modulate food intake, we surveyed the neuronal activation-like profile of *Cck-iCre*-hM3D$_q$-mCherry[NTS] mice treated with CNO or saline. Chemogenetic activation of CCK[NTS] neurons elicited striking c-Fos-IR in the paraventricular nucleus of the hypothalamus (PVH) in food-deprived mice compared with saline-treated mice (*Figure 3—figure supplement 1*). The PVH represents a crucial hub for the brain's regulation of energy balance and integrates a diverse range of nutritionally related hormonal and synaptic inputs, including inputs from the NTS (*Grill and Hayes, 2012*; *Riche et al., 1990*). To evaluate whether CCK[NTS] neurons directly innervate the PVH, a Cre-inducible AAV vector expressing Channelrhodopsin-2-eYFP (ChR2-eYFP) was stereotaxically injected into the NTS of *Cck-iCre* mice (*Figure 3A,B*). Cre-mediated recombination of the vector led to expression of eYFP in CCK[NTS] cells and their projection fields (due to axonal localization of the ChR2-eYFP fusion protein), revealing robust CCK[NTS] → PVH innervation (*Figure 3C*), with no substantial projections at other anterior hypothalamic regions (*Figure 3—figure supplement 2A–C*). CCK[NTS] efferent projections were also observed to innervate the ventral and more caudal aspect of the dorsomedial nucleus of the hypothalamus and, to a lesser extent, the arcuate nucleus (*Figure 3—figure supplement 2D–F*).

To probe for the physiological significance of this CCK[NTS] → PVH circuit, *Cck-iCre* mice injected into the NTS with AAVs encoding for ChR2-eYFP (a blue-light-sensitive cation channel) were implanted with an optic fiber above the PVH to stimulate these CCK[NTS] efferents (*Figure 3A–D*). *Ex*

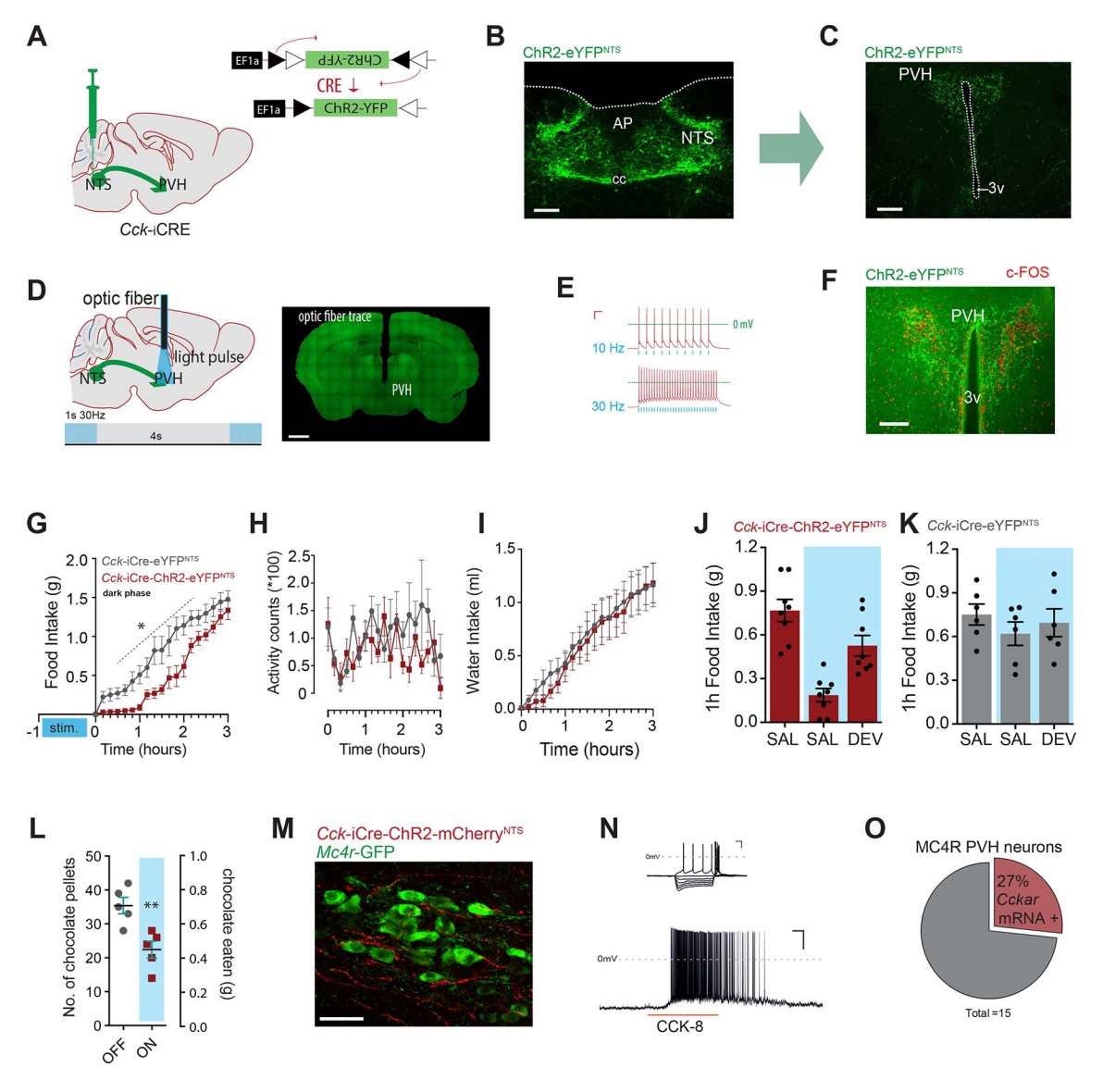

**Figure 3.** Activation of CCK[NTS] neurons efferent to the PVH suppresses appetite. (**A**) Schematic of CCK[NTS]→ PVH targeting strategy using bilateral NTS delivery of Cre-dependent ChR2-eYFP expressing vector in *Cck-iCre* mice. (**B**) Selective eYFP expression following Cre-mediated recombination in the caudal aspect of the NTS (scale bar represents 200 μm). (**C**) CCK[NTS] efferents (green) innervate the PVH (scale bar represents 400 μm). (**D**) CCK[NTS] axon targeting for photostimulation, positioning of the optic fiber and photostimulation parameters (scale bar represents 400 μm). (**E**) Current clamp recording of a CCK[NTS] neuron expressing ChR2 (scale bar 20 mV/100 ms). (**F**) Bilaterally transduced CCK[NTS] axons in the PVH and c-Fos-IR following PVH photostimulation (scale bar represents 200 μm). (**G**) In vivo optogenetic photostimulation of NTS[CCK]→PVH terminals in *Cck-iCre*-ChR2-eYFP[NTS] significantly reduced nocturnal feeding ($n = 6$, RM ANOVA: main effect of treatment ($F_{(1,10)} = 8.663$, $p = 0.0147$), main effect of time ($F_{(18,180)} = 97.25$, $p<0.0001$) and interaction ($F_{(18,180)} = 2.788$, $p = 0.0003$) Sidak's *post hoc* comparisons, *at least $p<0.05$ (**H**) without altering locomotor activity (RM ANOVA: main effect of treatment ($F_{(1, 10)} = 1.510$, $p = 0.2472$), main effect of time ($F_{(18,180)} = 1.797$, $p = 0.0285$) and interaction ($F_{(18,180)} = 1.198$, $p = 0.2671$) or (**I**) water consumption (RM ANOVA: main effect of treatment ($F_{(1,10)} = 0.0924$, $p = 0.7673$), main effect of time ($F_{(18,180)} = 50.68$, $p<0.0001$) and interaction ($F_{(18,180)} = 0.2666$, $p = 0.9989$) as compared to control *Cck-iCre*-eYFP[NTS]. Tick marks on x-axis represent 10-min intervals. (**J**) Real time food intake reduction following optogenetic activation of NTS[CCK]→PVH terminals in fasted *Cck-iCre*-ChR2-eYFP[NTS] and reversion following injection of CCK-A receptor antagonist (devazipide; DEV) ($n = 8$; RM one-way ANOVA, treatment $F_{(1.809,12.66)} = 16.15$, $p = 0.0004$; individual $F_{(7,14)} = 0.4241$, $p = 0.8714$, Sidak's *post hoc* comparison **$p<0.005$, *$p<0.05$). (**K**) Neither photostimulation nor DEV treatment alter food intake in fasted *Cck-iCre*-eYFP[NTS] control mice ($n = 6$; RM one-way ANOVA, treatment $F_{(1.294,6.469)} = 1.486$, $p = 0. 2780$; individual $F_{(5,10)} = 5.089$, $p = 0.014$). (**L**) Photostimulation of NTS[CCK]→PVH terminals reduces total intake of chocolate pellets over 30 min following 18–20 hr of food deprivation ($n = 5$, paired two-tailed *t*-test, $t_{(4)} = 6.949$, $p = 0.0023$), as compared to no photostimulation. (**M**) Representative confocal image of NTS[CCK]→PVH mCherry fibers and varicosities in close apposition to putative PVH MC4R-GFP neurons (scale bar represents 20 μm; 10 μm stack, maximum intensity projection). (**N**) Top, IV relationship

*Figure 3 continued on next page*

*Figure 3 continued*

of a PVH MC4R-GFP neuron produced by the superimposition of membrane potential deflection in response to a series of current injections of constant increment (scale bar 20 mV/200 ms); bottom, current clamp recording of the above neuron following bath application of CCK-8 (500 nM; scale bar 20 mV/30 s). (**O**) Percentage of PVH MC4R-neurons expressing CCK-A receptor mRNA as assessed by single-cell qPCR. NTS, nucleus of the solitary tract; PVH, paraventricular nucleus of the hypothalamus; AP, area postrema; cc, central canal; 3v, third ventricle.

The following figure supplements are available for figure 3:

**Figure supplement 1.** Chemogenetic activation of CCK[NTS] neurons elicits c-Fos in the paraventricular nucleus of the hypothalamus (PVH).

**Figure supplement 2.** Hypothalamic projections of CCKNTS neurons.

**Figure supplement 3.** Subset of PVH MC4R neurons express CCKA receptor and are responsive to CCK.

*vivo* application of blue light (473 nm) to CCK[NTS] neurons expressing ChR2 using slice electrophysiology resulted in faithful action potential discharge with stimulation/discharge fidelity preserved at both 10 Hz and 30 Hz (but not 50 Hz, data not shown) in *Cck*-iCre-ChR2-eYFP[NTS] cells (*Figure 3E*). In vivo photostimulation of CCK[NTS] axon terminals in the PVH elicited c-Fos-IR in PVH neurons in close proximity of transduced CCK[NTS] axons (*Figure 3F*).

We investigated whether specifically activating CCK[NTS] axon terminals within the PVH (10 ms, 30 Hz, 1 s on, every 4 s) prior the initiation of mouse nocturnal feeding could communicate a post-prandial-like signal, and thereby reduce subsequent spontaneous feeding. Monitoring home-cage food intake revealed that optogenetic activation of the CCK[NTS]→PVH circuit produced an almost complete, and reversible, suppression of spontaneous feeding (*Figure 3G*). Importantly, optogenetic activation of the CCK[NTS]→PVH pathway neither suppresses locomotor activity (*Figure 3H*) nor water consumption (*Figure 3I*). Furthermore, real-time optogenetic stimulation of this circuit (10 ms, 30 Hz, 1 s on, every 4 s) significantly suppressed re-feeding in food deprived *Cck*-iCre-ChR2-eYFP[NTS] mice (*Figure 3J*), but not control *Cck*-iCre-eYFP[NTS] mice (*Figure 3K*). The reduction in food intake in *Cck*-iCre-ChR2-eYFP[NTS] stimulated mice was blunted by pre-treatment with the CCK-A receptor antagonist devazepide (*Figure 3J*), which did not affect food intake in control mice (*Figure 3K*). Next, we tested the role of the CCK[NTS]→PVH circuitry to supress feeding in response to palatable food (chocolate pellets). Food deprived *Cck*-iCre-ChR2-eYFP[NTS] mice were offered chocolate pellets via a computer-controlled delivery system using a within-subjects design (optogenetic activation or no activation). Optogenetic activation of CCK[NTS]→PVH efferent reduced the total number of chocolate pellets consumed (*Figure 3L*), indicating that the appetite-suppressing properties of the CCK[NTS]→PVH circuit maintains salience even when the homeostatic drive to eat following food deprivation is boosted with a hedonic component.

Converging pharmacological and genetic data has established the importance of melanocortin-4 receptor (MC4R)-expressing neurons in the regulation of energy balance (*Fan et al., 1997*; *Huszar et al., 1997*; *Krashes et al., 2016*), with the PVH having been identified as a principal site of their satiety-promoting action (*Balthasar et al., 2005*; *Garfield et al., 2015*; *Shah et al., 2014*). To determine whether MC4R-expressing neurons are positioned to be a functional exponent of CCK[NTS]→PVH efferents, we first employed a modified bacterial artificial chromosome (BAC) transgenic MC4R-GFP reporter mouse line (*Ghamari-Langroudi et al., 2015*; *Liu et al., 2003*) and crossed it with the *Cck*-iCre line. CCK[NTS] neurons were transduced with a mCherry-conjugated ChR2 AAV vector in the *Cck*-iCre::MC4R-GFP double transgenic line. We observed CCK[NTS]→PVH mCherry-positive fibers and varicosities in close apposition to MC4R-expressing neurons (*Figure 3M* and *Figure 3—figure supplement 3*). Moreover, using patch clamp electrophysiology, we observed that approximately 23% (5/23 cells) of PVH MC4R-expressing neurons were excited by CCK-8 application (*Figure 3N*).

To increase the stringency of genetic identification of MC4R-expressing neurons, we also used a second and recently described knock-in mouse line (*Mc4r*-t2a-Cre) that express Cre recombinase from the endogenous and transcriptionally active *Mc4r* locus (*Garfield et al., 2015*) and in which MC4R-expressing neurons are fluorescently labeled by means of Cre-enabled tdTomato expression (*Mc4r*-t2a-Cre::tdTomato; *Figure 3—figure supplement 3A*). Single-cell qPCR of manually sorted

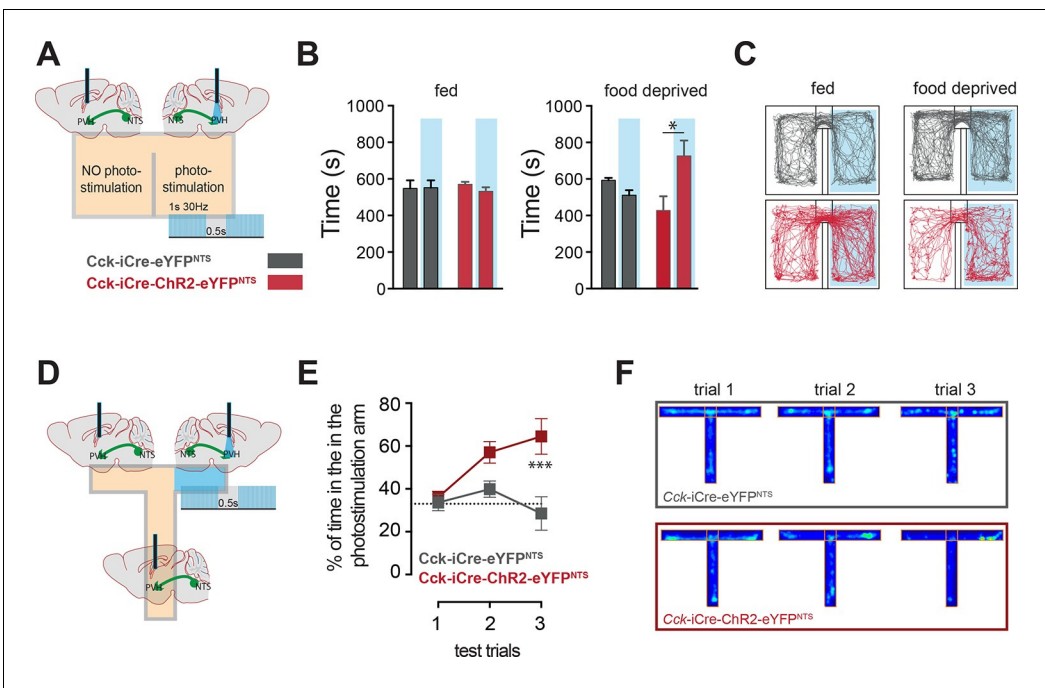

**Figure 4.** CCK[NTS] neurons signal positive valence via the PVH. (A and D) Experimental strategies for the interrogation of the valence encoded by NTS[CCK]→PVH terminals. (B) Food-deprived *Cck-iCre*-ChR2-eYFP[NTS] mice exhibited a significant place preference for the photostimulation-paired chamber during a real-time place preference assay, as compared to control *Cck-iCre*-eYFP[NTS] mice (n = 5–6; Two-way ANOVA – fed: no effect of photostimulation ($F_{(1,18)}$ = 0.3244, p = 0.5760), no effect of ChR2 ($F_{(1,18)}$ = 0.001145, p = 0.9734) or interaction ($F_{(1,18)}$ = 0.5007, p = 0.4883); food deprived: main effect of photostimulation ($F_{(1,18)}$ = 5.289, p = 0.0336), no effect of ChR2 ($F_{(1,18)}$ = 0.08811, p = 0.7700) and interaction ($F_{(1,18)}$ = 12.63, p = 0.0023); Sidak's *post hoc* comparisons, *p = 0.05). (C) Representative real-time place preference location plots one representative mouse per condition. (E) *Cck-iCre*-ChR2-eYFP[NTS] developed preference for the photostimulation-paired arm in a three-trial T-maze test (n = 5–6, main effect of treatment ($F_{(1,27)}$ = 15.93, p = 0.0005) main effect of trials ($F_{(2,27)}$ = 3.36, p = 0.0498) and interaction ($F_{(2,27)}$ = 4.36, p = 0.0228); Sidak's *post hoc* comparisons, ***p<0.001), as compared to *Cck-iCre*::eYFP[NTS]. (F) Representative T-maze location plots from one representative mouse per condition. NTS, nucleus of the solitary tract; PVH, paraventricular nucleus of the hypothalamus.

PVH *Mc4r*-t2a-Cre::tdTomato-expressing neurons revealed *Cckar* mRNA to be expressed in 27% (4/15) of MC4R PVH cells analyzed (*Figure 3O* and *Figure 3—figure supplement 3C*). Likewise, electrophysiological recordings revealed that approximately 30% (6/20 cells) of PVH *Mc4r*-t2a-Cre::tdTomato neurons were excited by CCK-8 application, in a CCK-A receptor-dependent manner (0/11 cells; *Figure 3—figure supplement 3D*). Thereby, these data identify CCK as a novel peptide neurotransmitter activating the appetite-controlling PVH MC4R neurons.

## A CCK[NTS]→PVH circuit encodes positive valence

In addition to the homeostatic regulation of energy balance, the NTS is also associated with negative valence and anorexia related to nausea (*Lachey et al., 2005*; *Rinaman, 2004*; *Swank and Bernstein, 1994*). We next considered whether the CCK[NTS]→PVH circuit is associated with the induction of an aversive physiological state. The motivational valence of this circuit was assessed using a real-time place preference test (*Stamatakis and Stuber, 2012*). We observed no place preference when *Cck*-iCre-ChR2-eYFP[NTS] or *Cck*-iCre-eYFP[NTS] controls were tested under normal energy balance conditions, an important indication that activation NTS[CCK]→PVH efferents is not aversive (*Figure 4A–C*). However, while *Cck*-iCre-eYFP[NTS] controls maintained no chamber preference when food deprived, calorie depleted *Cck*-iCre-ChR2-eYFP[NTS] mice exhibited a significant preference for the photostimulation-paired chamber (*Figure 4B,C*). This behavioral response is reminiscent of that observed following direct optogenetic activation of PVH MC4R-expressing neurons (*Garfield et al., 2015*). We

further examined this observation using a T-shaped maze. In this test, exploration of one of the three maze's arms was paired with the photostimulation of the NTS$^{CCK}$→PVH efferent (*Figure 4D*). Over the three testing trials *Cck*-iCre-ChR2-eYFP$^{NTS}$ mice, but not *Cck*-iCre-eYFP$^{NTS}$ controls, developed a clear preference for the photostimulation-paired arm (*Figure 4E,F*). Thus, when hungry and in the absence of food, mice sought out activation of the NTS$^{CCK}$ efferent to PVH, revealing the CCK$^{NTS}$→PVH circuit encodes positive valence. These data suggest that the activation of this circuit provides relief from the unpleasantness of energy deficit by mimicking a post-prandial phase.

While further studies attempting site-specific *Cck* loss of function are needed to fully clarify the physiological necessity of NTS CCKergic transmission in eating behavior and body weight regulation, here we reveal that activation of CCK$^{NTS}$ produces a prolonged effect on appetite and a rapid reduction in body weight. We did not find evidence that this reduction in energy intake is due to an aversive state, but rather, reveal that the CCK$^{NTS}$→PVH circuit transmits positive valence in an energy-state-dependent manner; a particularly attractive prospect considering that in patient populations the negative valence of energy deficit is a major factor contributing to low compliance on weight-loss diets.

## Materials and methods

### Animals

CCK-ires-Cre (*Cck*-iCre; *Cck$^{tm1.1(cre)Zjh}$*/J; Stock No. 012706), R26-loxSTOPlox-eYFP (Ai3; B6.Cg-Gt (ROSA)26Sor$^{tm3(CAG-EYFP)Hze}$/J; Stock No. 007903), MC4R-GFP (B6.Cg-Tg(Mc4r-MAPT/Sapphire) 21Rck/J; Stock No. 008323) or R26-loxSTOPlox-tdTomato (Ai9; B6.Cg-Gt(ROSA)26Sor$^{tm9(CAG-tdTomato)Hze}$/J; Stock No.007909) were obtained from Jackson Laboratories (Bar Harbor, ME). *Mc4r-t2a-Cre* mice were previously described (*Garfield et al., 2015*). Mice were provided with standard mouse chow and water *ad libitum*, unless otherwise noted and housed at 22–24°C with a 12-hr light/ 12-hr dark cycle. All experimental procedures were performed in accordance with the UK Animals (Scientific Procedures) Act 1986 or the Beth Israel Deaconess Medical Center Institutional Animal Care and Use Committee.

### Drugs

Drugs for in vivo use were prepared in sterile saline and administered intraperitoneally (IP). Clozapine-N-oxide (CNO; Tocris Bioscience, Bristol, UK; Cat. No. 4936) was administered at 0.3 mg/kg. Devazepide (Tocris Bioscience, Cat. No. 2304) was dissolved in DMSO, further diluted with sterile saline and administered at 1 mg/kg, IP, 40 min before CNO injection or optogenetic photostimulation. CCK-8 (Tocris, Cat. No. 1166) and SR 27,897 (CCK-A antagonist; Tocris Bioscience, Cat. No. 27897) were dissolved in artificial cerebrospinal fluid (aCSF).

### Viral vectors

DREADD and Optogenetics viruses employed were obtained from University North Carolina Vector Core (Chapel Hill, NC). DREADD constructs were packaged in AAV serotype-8 and injected at a titer of 1.3 × 10$^{12}$ vg/ml. AAV-EF1a-DIO-EYFP, AAV/EF1a-DIO-ChR2(E123T/T159C)-eYFP, AAV/EF1a-DIO-ChR2(E123T/T159C)-mCherry were packaged in AAV serotype-2 and injected at a titer of about 1 × 10$^{12}$ vg/ml. Nucleus-specific delivery of AAVs was achieved through stereotaxic injections.

### Stereotaxic surgery

NTS delivery of AAVs was achieved through a modified stereotaxic procedure. Briefly, 5–8 week-old male mice were anesthetized with a mixture of ketamine and xylazine dissolved in saline (100 and 10 mg/kg, respectively; 10 ml/kg). Mice were placed in a stereotaxic frame (Kopf Instruments) using ear bar with head angled at about 45°. Under magnification, an incision was made at the level of the *cisterna magna* and neck muscles carefully retracted. A 33G needle was used for *dura* incision. The *obex* served as reference point for injection. Injections were performed using a glass micropipette (diameter 20–40 μm). NTS coordinates were A/P, -0.2; M/L, ±0.2; D/V, -0.2 from the *obex*. Virus was delivered under air pressure using a PLI-100A Pico-Injector (Harvard Apparatus, Cambridge, UK). About 150 nl of virus/side were delivered with multiple microinjections over 4–5 min. The pipette remained in place for a minimum of 3 min after injection. Animals were administered an analgesic (5

mg/kg Carprofen, subcoutaneously) for 2 days post-operatively and given a minimum of 14 days recovery before being used in any DREADD experiments. During this time, mice were acclimated to handling and IP injections. For optogenetic experiments, optic fibers were placed above the PVH at least 4 weeks after the NTS viral delivery to ensure labelling of distal projection sites. Based on the Franklin and Paxinos Mouse Brain Atlas (*Franklin and Paxinos, 2008*), the following coordinates were used for targeting optic fibers (200 µm diameter core; CFMLC22U-20; Thorlabs, Newton, NJ) at the PVH (mm from Bregma): A/P, −0.65; M/L, ± 0; D/V, −4.20 mm. After placement of the optic fiber, mice were allowed additional 14 days post-surgical recovery, as described above. Mice were excluded from behavioral analysis if *post hoc* histological validation showed either no viral transduction or misplaced optical fibers.

## Energy balance and body weight studies

*Dark-cycle food intake.* Mice were injected with saline or CNO (0.3 mg/kg; IP Tocris Bioscience) 30 min prior to the onset of the dark. At onset of dark, food was returned and intake recorded automatically via the TSE Phenomaster system (TSE, Bad Homburg, Germany) or manually (*Figure 2g*). *Post-fast re-feed.* Mice were fasted overnight. The following morning, mice were treated IP with saline or CNO and food returned 30 min later and intake recorded manually. *Daily treatment.* Mice were treated IP twice daily for 4 days with 0.3 mg/kg of CNO or saline near the onset of the dark and light cycles. Food intake and body weight were recorded daily prior to the dark cycle. *Nutrient gavage.* Mice were fasted overnight and the following morning, received intragastric 0.5 ml volume gavage of water, amino acids (Peptone enzymatic digest from casein, Sigma-Aldrich, Dorset, UK; Cat. No. 70172) or sucrose (Sigma-Aldrich, Cat. No. 84100). Amino acids and sucrose were diluted in drinking water and delivered in an amount of 1kcal. *Blood glucose studies.* Mice were food deprived overnight and blood glucose concentration was determined from tail bleeds using OneTouch Ultra glucometer and test strips (LifeScan, Johnson and Johnson Medical Limited, Livingstone, UK). Basal blood glucose concentration was determined prior to injection of any substances.

## Photostimulation protocol

Fiber optic cables (1.5 m long, 200 µm diameter, 0.22 NA; Doric Lenses, Franquet, Quebec, Canada) were firmly attached to the implanted fiber optic cannulae with zirconia sleeves (Doric Lenses). Photostimulation was programmed using a pulse generator software (Prizmatix, Givat-Shmuel, Israel) that controlled a blue light laser (473 nm; Laserglow, Toronto, Canada) via a USB-TTL interface (Prizmatix).

Photostimulation for feeding experiments: light pulse trains 10-ms pulse width, 30 Hz, 1 s on, 4 s off. Photostimulation for place preference and T-maze tests: light pulse trains 10-ms, 30 Hz, 1 s on, 0.5 s off. Light power was adjusted such that the light power exiting the fiber optic cable was at least 10 mW using a digital optical power meter (PM100D, Thorlabs) and an online light transmission calculator for brain tissue (http://web.stanford.edu/group/dlab/cgi-bin/graph/chart.php). After the completion of photostimulation experiments, mice were perfused and the approximate locations of fiber tips were identified according to the atlas coordinates (*Franklin and Paxinos, 2008*). After removal, optic fiber were connected to optic fiber cable and tested for light transmission.

## Behavioral testing

Mice were tested for real-time place preference in a model in which one chamber was paired with 30-Hz photostimulation and the other, identical chamber resulted in no photostimulation. Total test duration was 20 min. A similar protocol was used for the T-maze experiments. For the T-maze test mice were tested in three 10-min trials with an inter-trail interval of 15–20 min, during which mice were returned to the home cage. Time spent in the photostimulation versus non-photostimulation zones was recorded via a CCD camera interfaced with Any-maze software (Stoelting, Wood Dale, IL).

## Brain tissue preparation and immunohistochemistry

Following deep terminal anesthesia with pentobarbital, mice were transcardially perfused with phosphate-buffered saline (PBS) followed by 10% neutral buffered formalin (Fisher Scientific, Loughborough, UK). Brains were extracted, cryoprotected in 30% sucrose in PBS,

sectioned coronally on a freezing sliding microtome (Bright solid state freezer series 8000, Bright Instruments, Luton, UK) at 30 µm and collected in four equal series. IHC was performed using standard methods and as previously described (*Garfield et al., 2012*). Briefly, sections were washed in PBS before blocking in 0.5% BSA/0.25% Triton X-100 in PBS for 1 hr at room temperature. Tissue was incubated overnight at room temperature in blocking buffer containing the primary antibodies: rabbit anti-c-FOS (EMD Millipore, Livingston, UK; Cat. No. PC05; diluted 1/5000), or chicken anti-GFP (Abcam, Cambridge, UK; Cat. No. ab13970; diluted 1/1000). The next day, sections were washed in PBS then incubated in blocking buffer containing appropriate secondary antibody (Alexa Fluor, Life Technologies; diluted 1/500) for 1 hr. 3,3′-Diaminobenzidine (DAB) staining was used for used for detection of c-Fos-IR using: a rabbit anti-c-FOS (EMD Millipore Cat. No. PC05; diluted 1/5000), secondary biotin-SP donkey anti-rabbit (Jackson Immunoresearch, West Grove, PA; Cat. No. 711-065-152; diluted 1/250), an avidin/biotin-based peroxidase system (Vectastain Elite ABC Kit, Vector Laboratories, Burlingame, CA; Cat. No. PK-6100) and a developing kit (DAB Peroxidase Substrate Kit, Vector Laboratories, Cat. No. SK-4100). c-Fos-IR was pseudocolored using Photoshop software to appear colored in images. Sections were mounted onto microscope slides and cover-slipped in an aqueous mounting medium (Vectashield Antifade Mounting Medium, Vector Laboratories, Cat. No. H-1000). Slides were imaged on a VS120 slide scanner (Olympus, Essex, UK) or AXIOSKOP2 (Zeiss, Oberkochen Germany). For counting of c-FOS-IR nuclei, the boundaries of the nucleus were defined using neuroanatomical landmarks and the Franklin and Paxinos Mouse Brain Atlas (*Franklin and Paxinos, 2008*).

## Electrophysiological studies

For electrophysiological validation mice were anesthetized with sodium pentobarbital (Euthatal) and decapitated. The brain was rapidly removed and placed in cold (0–4°C), oxygenated (95%$O_2$/5% $CO_2$) 'slicing' solution containing (in mM) sucrose (214), KCl (2.5), $NaH_2PO_4$ (1.2), $NaHCO_3$ (26), $MgSO_4$ (4), $CaCl_2$ (0.1), D-glucose (10). The brain was glued to a vibrating microtome (Campden Instruments, Loughborough, UK) and 200-µm thick coronal sections of the brainstem containing the NTS were prepared. Slices were immediately transferred to a' recording' solution containing (in mM) NaCl (127), KCl (2.5), $NaH_2PO_4$ (1.2), $NaHCO_3$ (26), $MgCl_2$ (1.3), $CaCl_2$ (2.4), D-glucose (10), in a continuously oxygenated holding chamber at 35°C for a period of 25 min. Subsequently, slices were allowed to recover in 'recording' solution at room temperature for a minimum of 1 hr before recording. For whole-cell recordings, slices were transferred to a submerged chamber and a Slicescope upright microscope (Scientifica, Uckfield, UK) was used for infrared - differential interference contrast and fluorescence visualization of cells. During recording slices were continuously perfused at a rate of ca. 2 ml/min with oxygenated 'recording' solution (as above) at room temperature. All pharmacological compounds were bath applied. Whole cell current-clamp recordings were performed with pipettes (3–7 MΩ when filled with intracellular solution) made from borosilicate glass capillaries (World Precision Instruments, Aston, UK) pulled on a Zeitz DMZ micropipette puller (Zeitz Instruments GmBH, Martinsried, Germany). The intracellular recording solution contained (in mM) K-gluconate (140), KCl (10), HEPES (10), EGTA (1), $Na_2ATP$ (2), pH 7.3 (with KOH). Recordings were performed using a Multiclamp 700B amplifier and pClamp10 software (Molecular Devices, Sunnyvale, CA). Liquid junction potential was 16.4mV and not compensated. The recorded current was sampled at 10 kHz and filtered at 2 kHz unless otherwise stated. Photostimulation of channelrhodopsin2 was achieved by 470 nm blue light delivered via the microscope objective. Light was generated by a pE-4000 LED illumination system (CoolLED, Andover, UK) driven via clampex 10.4 software (Molecular Devices, Sunnyvale, CA). For CCK-8 application on MC4R PVH neurons, brain coronal slices from 6–8 week-old *Mc4r*-GFP or *Mc4r*-t2a-Cre::tdTomato mice were prepared. Extracted brains were immediately submerged in ice-cold, carbogen-saturated (95% O2, 5% CO2) high-sucrose solution (238 mM sucrose, 26 mM NaHCO3, 2.5 mM KCl, 1.0 mM NaH2PO4, 5.0 mM MgCl2, 10.0 mM CaCl2, 11 mM glucose). Then, 300-µm thick coronal sections were cut with a Leica VT1000S vibratome and incubated in oxygenated aCSF (126 mM NaCl, 21.4 mM NaHCo3, 2.5 mM KCl, 1.2 mM NaH2PO4, 1.2 mM MgCl2, 10 mM glucose) at 34°C for 30 min. Then, slices were maintained and recorded at room temperature (20–24°C). The intracellular solution for current clamp recordings contained the following (in mM): 128 K gluconate, 10 KCl, 10 HEPES, 1 EGTA, 1 MgCl2, 0.3 CaCl2, 5 Na2ATP, 0.3 NaGTP, adjusted to pH 7.3 with KOH. CCK-8 (100 nM) and SR 27,897 (CCK-A

antagonist; 250 nM) were applied to the bath through perfusion. Synaptic blockers (1 mM kynuerenate and 100 µM picrotoxin) were added to the aCSF to synaptically isolate MC4R PVH neurons.

## Single-cell qPCR

Adult (4–5 week old) *Mc4r*-t2a-Cre::tdTomato male mice (n=2) were anesthetized with isoflurane. Brains were extracted and immediately chilled in ice-cold, carbogen-saturated (95% $O_2$, 5% $CO_2$) high-sucrose solution (238 mM sucrose, 26 mM $NaHCO_3$, 2.5 mM KCl, 1.0 mM $NaH_2PO_4$, 5.0 mM $MgCl_2$, 10.0 mM $CaCl_2$, 11 mM glucose). Next, 300-µm thick coronal sections were cut with a Leica VT1000S Vibratome and incubated in oxygenated aCSF (126 mM NaCl, 21.4 mM $NaHCO_3$, 2.5 mM KCl, 1.2 mM $NaH_2PO_4$, 1.2 mM $MgCl_2$, 2.4 mM $CaCl_2$, 10 mM glucose) at 34°C for 30 min. The PVH was visualized by fluorescence stereoscope then micro-dissected and enzymatically dissociated according to a published protocol (*Saxena et al., 2012*), except that trituration was performed with fire-polished Paster pipettes. From the resulting cell suspension, tdTomato+ cells were individually washed and collected (*Hempel et al., 2007*), frozen at -80°C and then processed for cDNA synthesis and amplification (*Picelli et al., 2014*). To control for mRNA contamination during cell collection, an equivalent volume of cell-picking buffer was sampled and processed along with each batch of cell samples. After 20 cycles of amplification by polymerase chain reaction (PCR), cDNA was purified (*Picelli et al., 2014*) and eluted in 30 µl PCR-grade water, and then analyzed by quantitative PCR (qPCR) for *Actb, tdTomato,* and *Cckar* in duplicate reactions. *Actb* and *Cckar* qPCR assays were obtained from Integrated DNA Technologies (IDT, Coralville, IA; Cat. No. Mm.PT.58.28904620.g and Mm.PT.58.12665706IDT). The *tdTomato* assay was custom synthesized by IDT from the following sequences: left, ACCCAGACCGCCAAGCTGAA; right primer, AGTTCATCACGCGCTCCCACT; internal probe, GCCCCCTGCCCTTCGCCTGG.

## Statistics

Statistical analyses were performed using Prism 6 (Graphpad Software, La Jolla, CA). Data were analyzed using t-test, one-way ANOVA, two-way or repeated measures (RM) ANOVA with *post hoc* comparisons, where appropriate. *N* represents independent biological replicates. No statistical methods were used to predetermine sample sizes. Sample size was computed based on pilot data and published literature. Data are presented as mean ± SEM and statistical significance was set at p<0.05.

## Acknowledgements

Authors wish to thank members of staff of the Medical Research Facility (University of Aberdeen) for assisting with mouse care and husbandry and Raffaella Chianese for assisting with mouse genotyping. This work was supported by the Wellcome Trust (LKH; WT098012), Biotechnology and Biological Sciences Research Council (LKH and GD'A; BB/K001418/1), Wellcome Trust and the University of Aberdeen (GD'A; 105625/Z/14/Z), NIDDK (BBLo; R01 DK075632, P30 DK046200, P30 DK 057521) and American Heart Association (JNC; 14POST20100011). Single-cell qPCR experiment was supported in part by the Molecular Medicine Core facility at Beth Israel Deaconess Medical Center.

## Additional information

### Funding

| Funder | Grant reference number | Author |
| --- | --- | --- |
| Wellcome Trust | WT098012 | Lora K Heisler |
| Biotechnology and Biological Sciences Research Council | BB/K001418/1 | Giuseppe D'Agostino Lora K Heisler |
| Wellcome Trust | 105625/Z/14/Z | Giuseppe D'Agostino |
| National Institute of Diabetes and Digestive and Kidney Diseases | R01 DK075632 | Bradford B Lowell |

| National Institute of Diabetes and Digestive and Kidney Diseases | P30 DK046200 | Bradford B Lowell |
| --- | --- | --- |
| National Institute of Diabetes and Digestive and Kidney Diseases | P30 DK 057521 | Bradford B Lowell |
| American Heart Association | 14POST20100011 | John N Campbell |

The funders had no role in study design, data collection and interpretation, or the decision to submit the work for publication.

## Author contributions

GD, Conception and design, Acquisition of data, Analysis and interpretation of data, Drafting or revising the article; DJL, Contributed electrophysiological recordings, Acquisition of data, Analysis and interpretation of data, Drafting or revising the article; CC, LKB, APG, Helped to perform all other experiments, Acquisition of data, Analysis and interpretation of data, Drafting or revising the article; JCM, Contributed electrophysiological recordings, Contributed reagents/analytic tools, Acquisition of data, Analysis and interpretation of data, Drafting or revising the article; JNC, Contributed single-cell qPCR experiments, Contributed reagents/analytic tools, Acquisition of data, Analysis and interpretation of data, Drafting or revising the article; BBLa, Helped to perform all other experiments, Analysis and interpretation of data, Drafting or revising the article; BBLo, Contributed reagents/analytic tools, Analysis and interpretation of data, Drafting or revising the article, Contributed unpublished essential data or reagents; RJD, Analysis and interpretation of data, Drafting or revising the article; LKH, Conception and design, Analysis and interpretation of data, Drafting or revising the article

## Author ORCIDs

Claudia Cristiano, http://orcid.org/0000-0002-4806-1764
Lora K Heisler, http://orcid.org/0000-0002-7731-1419

## Ethics

Animal experimentation: All experimental procedures were performed in accordance with the UK Animals (Scientific Procedures) Act 1986 (Project License No. 60/4565).

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
