## [Decision Letter]

Thank you for submitting your work entitled "Appetite controlled by a cholecystokinin nucleus of the solitary tract to hypothalamic neurocircuit" for consideration by *eLife*. Your article has been reviewed by three peer reviewers, and the evaluation has been overseen by a Reviewing Editor (Joseph Takahashi) and a Senior Editor.

The following individual involved in review of your submission has agreed to reveal their identity: Joseph Bass (peer reviewer).

The reviewers have discussed the reviews with one another and the Reviewing Editor has drafted this decision to help you prepare a revised submission.

Summary:

This manuscript submitted as a short report by Heisler and colleagues represents an elegant advance in the genetic dissection of the neurocircuitry underlying satiety and its connection to long term appetitive and metabolic control. While the wave of studies applying functional genetic analysis to feeding have predominately focused on canonical leptin-responsive neurons within the ARN, including NPY/AgRP and POMC-expressing cells, much less has been examined regarding the complex organization and mechanisms involving cell groups within brainstem. A gap has remained in understanding how long-term (leptin responsive) regions communicate with acute-response centers in brainstem, and the mechanisms of peripheral-to-central communication (i.e., gut-to-brain signaling) that transduce incretin response – both topics that are illuminated in this work. A long line of investigation has implicated the gut-peptidergic molecule CCK as a key regulator of satiety, and the present work provides the first analysis of the pathways involved at the genetic, molecular, and behavioral levels. Using a genetically targeted fluorescent probe, CCK expressing neurons within the NTS are identified as nutrient state dependent, being activated by gastric gavage with nutrient but not saline. Chemogentic testing reveals that activation of CCK-NTS cell bodies results in both inhibition of refeeding response to 24 h fasting, and, with chronic stimulation, weight loss that is rapidly recovered following cessation of the stimulation. Pharmacologic antagonism of CCK abrogated the effects genetic stimulation of these neurons, consistent with mediation via CCK-R signaling as the mediator of the NTS-derived stimulus. At an anatomic level, data is provided using genetically-encoded cell type-specific tracing to establish the NTS->PVN distribution track as a pathway for CCK-induced signaling, and optogenetic approaches demonstrate that activation of CCK-responsive PVN cells leads to inhibition of the homeostatic fasting-refeeding response to food deprivation through a mechanism corresponding with activation of closely juxtaposed MCR4-expressing cells on patch clamp analysis. Additional experiments indicate that effects of CCK cannot be attributed to aversive properties, consistent with the "positive valence" of the CCK-target PVN cells on behavioral testing. At the physiologic level, past work by these investigators and their collaborators has established a role for leptin in the tonic regulation of CCK-responsive neurons in the parabrachial nucleus, revealing cross-talk between the adipostatic hormone leptin and CCK in the counterregulatory response to glucodeprivation. That adipostatic (leptin) and satiety (CCK) signals are anatomically and physiologically integrated/convergent at the level of brainstem and PVN is now firmly established in the present work.

As such, the findings are of broad interest in both neuroscience and metabolism not only for advancing insight into the neurocircuitry processing appetitive and behavioral responses to energy state, but also for expanding our understanding of the mechanisms underlying communication (and the "gateway") between peripheral tissues and brain. The work as presented is both polished and detailed, with only a few points that arise for clarification.

Essential revisions:

1) Please address or respond to the following criticism: The largest concern of the manuscript is that it uses only gain-of-function approaches. A growing body of evidence points to both DREADD and optogenetics ability to turn on circuits in a way that would never happen in a normal context. Loss of function approaches to this pathway would have the distinct advantage of proving that the normal activation of this circuit would regulate food intake. This should be addressed in the text of the revised manuscript.

---

## [Author Response]

Essential revisions: 1) Please address or respond to the following criticism: The largest concern of the manuscript is that it uses only gain-of-function approaches. A growing body of evidence points to both DREADD and optogenetics ability to turn on circuits in a way that would never happen in a normal context. Loss of function approaches to this pathway would have the distinct advantage of proving that the normal activation of this circuit would regulate food intake. This should be addressed in the text of the revised manuscript.

The primary objective of the work reported here was to identify a circuit that may ultimately have translational benefit for obesity treatment through exogenous (pharmacological) stimulation. Thus, we adopted exogenous stimulation strategies, DREADD and optogenetics, and we uncovered a new CCK^NTS^→PVH circuit that when engaged has a profound impact on food intake and body weight. A different question is the role of normal circulating CCK^NTS^ in ingestive behavior and body weight regulation. Future studies using site-specific CCK loss of function will determine the necessity of endogenous CCK^NTS^ in the regulation of food intake and body weight. To address this point, we now clarify in the revised manuscript:

“While further studies attempting site-specific *Cck* loss of function are needed to fully clarify the physiological necessity of NTS CCKergic transmission in eating behavior and body weight regulation, here we reveal that activation of CCK^NTS^ produces a prolonged effect on appetite and a rapid reduction in body weight.”